# A Reconfigurable Hardware Architecture for Miscellaneous Floating-Point Transcendental Functions

**Peng Li** [1,†], **Hongyi Jin** [2,†], **Wei Xi** [1], **Changbao Xu** [3], **Hao Yao** [1] **and Kai Huang** [2,*]

1   Digital Grid Research Institute, China Southern Power Grid, Guangzhou 510670, China
2   School of Micro-Nano Electronics, Zhejiang University, Hangzhou 310030, China
3   Electric Power Research Institute, Guizhou Power Grid Co., Ltd., Guiyang 550002, China
*   Correspondence: huangk@zju.edu.cn
†   These authors contributed equally to this work.

**Abstract:** Transcendental functions are an important part of algorithms in many fields. However, the hardware accelerators available today for transcendental functions typically only support one such function. Hardware accelerators that can support miscellaneous transcendent functions are a waste of hardware resources. In order to solve these problems, this paper proposes a reconfigurable hardware architecture for miscellaneous floating-point transcendental functions. The hardware architecture supports a variety of transcendental functions, including floating-point sine, cosine, arctangent, exponential and logarithmic functions. It adopts the method of a lookup table combined with a polynomial computation and reconfigurable technology to achieve the accuracy of two units of least precision ($ulp$) with 3.75 KB lookup tables and one core computing module. In addition, the hardware architecture uses retiming technology to realize the different operation times of each function. Experiments show that the hardware accelerators proposed can operate at a maximum frequency of 220 MHz. The full-load power consumption and areas are only 0.923 mW and $1.40 \times 10^4$ $\mu m^2$, which are reduced by 47.99% and 38.91%, respectively, compared with five separate superfunction hardware accelerators.

**Keywords:** floating-point transcendental functions; reconfigurable; lookup table; polynomial

## 1. Introduction

Transcendental functions refer to functions that cannot be represented by finite quadric operations, power operations or square root operations, such as trigonometric functions, inverse trigonometric functions, exponential functions and logarithmic functions. They are basic components of mathematical calculations and are widely used in algorithms in various fields [1].

For some data-intensive algorithms with strict real-time requirements, the low-latency computation of transcendental functions is of great significance. A large number of intensive floating-point trigonometric functions, exponential and logarithmic operations are usually required in the fields of motor control, noise filtering, digital signal processing, etc. [2–6]. In the field of electric power, trigonometric functions are widely used in the calculation of power quality, complex harmonic processing and phase calculation [7]. It takes a lot of cycles for the software program to perform those function operations, which cannot meet real-time requirements. Therefore, a low-latency high-precision transcendental-function hardware accelerator is needed. Furthermore, multiple transcendental functions are frequently required to be operated in some high-complexity algorithms. These transcendental functions require a lot of hardware resources when implemented separately, resulting in large area overheads [8,9]. Therefore, a hardware accelerator that supports multiple transcendental functions is of great significance.

Researchers have also already proposed various hardware accelerators to implement transcendental functions' calculation, including the CORDIC algorithm [10–12] and the

piecewise polynomial approximation method [13–15]. However, the CORDIC algorithm requires multiple iterations to achieve a high accuracy, and it supports only limited transcendental functions [16–18]. Although the lookup table method combined with polynomial operation can easily and effectively implement a transcendental function [19–24], there are often multiple data paths when implementing multiple transcendental functions, resulting in a waste of hardware resources.

A reconfigurable hardware architecture for miscellaneous floating-point transcendental functions is proposed in this paper. High-precision lookup tables for polynomial coefficients of transcendental functions are generated by polynomial fitting. The whole calculation is divided into preprocessing, core computing and postprocessing to achieve a low-latency hardware design. With reconfigurable technology, multiple transcendental function calculations can be implemented by the same core module, which leads to a lower area cost of hardware.

This paper identifies two key challenges in designing a reconfigurable low-latency hardware architecture for floating-point transcendental functions. Firstly, how to fit each transcendental function using a piecewise lookup table combined with a polynomial computation to obtain a high-precision polynomial coefficient lookup table in order to achieve a low-latency hardware architecture. Secondly, how to design a core computing unit that can implement multiple transcendental functions in the smallest possible area while maintaining a high accuracy.

The current research cannot achieve the balance of low delay and high precision for transcendental functions. For example, ref. [25] had high accuracy but a high computation delay. Its error was no more than 1.5 $ulp$, and its delay was 15 cycles. In contrast, ref. [26] only needed to spend 40.3 ns to compute the transcendental function with a maximum error of $1 \times 10^{-9}$. In the present research, different transcendental functions have different implementation processes, so they cannot be implemented using the same data path.

The reconfigurable hardware architecture for miscellaneous transcendental function proposed in this paper uses only 3.75 KB of lookup tables to implement five high-precision floating point transcendental functions, including sine, cosine, arctangent, exponential and logarithmic functions. The difference between the calculation result of the hardware circuit and that of the C language math library is most 2 $ulp$. Under the UMC 40 nm CMOS process, the hardware can reach a maximum frequency of 220 MHz. The synthesis results also show that the total area is $1.40 \times 10^{4}$ μm$^2$ and the full-load power consumption is 0.923 mW.

## 2. Related Work

The CORDIC algorithm is generally used to achieve transcendental functions in digital circuit design. Muñoz et al. [27] used the CORDIC algorithm and a Taylor series expansion to calculate sine, cosine and arctangent functions, which implied floating-point operations and a search in ROM method to achieve a high throughput. Sergiyenko et al. [25] applied the CORDIC algorithm to achieve transcendental functions in three stages, whose angles were, respectively, from the ROM table, a network of CORDIC microrotations and an approximation network, so as to minimize the area and delay. It only took 15 cycles to achieve an accuracy of 0.5 $ulp$ for the sine and cosine calculations of a small angle.

In addition to the CORDIC algorithm, a lookup table method combined with a polynomial computation is also an important approach to implementing transcendental functions. Chen et al. [13] proposed a logarithmic function hardware accelerator based on lookup tables with 7.8 KB of lookup tables and a large number of basic computing units, achieving an accuracy of 3.5 $ulp$ and a latency of 78 ns. Gener et al. [28] presented a lossless LUT compression method which could be used to replace tables among other applications of LUTs. Their method resulted in a 10% performance improvement, but only two transcendental functions were supported in that work. The hardware unit of the high-speed transcendental function proposed by Tian et al. [14] used a binomial operation, and its accuracy reached $1 \times 10^{-7}$. However, the multiple data paths that were necessary to implement multiple

transcendental functions resulted in an excessive area overhead. Nandagopal et al. [15] proposed a novel piecewise-linear method to approximately represent nonlinear logarithmic and antilogarithmic functions. In that study, the calculation delay reached 15.20 ns, but the accuracy could only reach $5 \times 10^{-6}$.

In summary, the iterative CORDIC algorithm for transcendental functions is very inferior in performance. It needs to spend more clock cycles to complete a single-precision floating-point operation with high accuracy. The lookup table method combined with a polynomial computation can achieve single-precision floating-point operations with low latency by using a small amount of storage and hardware resources.

## 3. Methods

The polynomial algorithm of single-precision floating-point transcendental functions based on the lookup table method has the following main steps:

1. Reduce the input range of the transcendental function to the convergence interval;
2. Select an appropriate polynomial to fit it and determine the values of the polynomial parameters;
3. Hardware implementation.

### 3.1. Preliminary Range Reduction

According to the IEEE-754 standard, a 32-bit normalized single-precision floating-point number consists of one signal bit $S$, an 8-bit exponent $E$ and a 23-bit mantissa $T$, which can be represented by Equation (1). Moreover, the range of normalized single-precision floating-point values is $(-2^{128}, -2^{-126}] \cup [2^{-126}, 2^{128})$ [29,30].

$$x = (-1)^S \times 2^{E-127} \times (1 + \frac{T}{2^{23}}) \tag{1}$$

The expression of the sine function implemented in this paper is Equation (2).

$$y = \sin \pi x, x \in (-\infty, +\infty) \tag{2}$$

The sine function has periodicity and symmetry, so any input can be transformed into a new input within the interval $[0, 0.5]$. It can be transformed into Equation (3).

$$y = \sin \pi x, x \in [0, 0.5] \tag{3}$$

The inputs of the cosine function, like the sine function inputs, can be transformed into a new input within the interval $[0, 0.5]$ for an operation. Moreover, it can be implemented using the sine function via Equation (4).

$$y = \cos \pi x = \sin ((0.5 - x)\pi), x \in [0, 0.5] \tag{4}$$

The expression of the arctangent function implemented in this paper is Equation (5) [31,32].

$$y = \arctan (x)/\pi, x \in (-\infty, +\infty) \tag{5}$$

The arctangent function is an odd function that can transform any input into the interval $[0, +\infty)$ for an operation. For inputs in the interval $[1, +\infty)$, they can be computed indirectly using a transformation, as shown in Equation (6). Therefore, the range of inputs of the arctangent function can be reduced to the interval $[0, 1]$.

$$y = \arctan (x)/\pi = 0.5 - arctan(\frac{1}{x})/\pi, x \in [1, +\infty) \tag{6}$$

The expression of the exponential function implemented in this paper can be transformed into Equation (7):

$$y = e^x = 2^{x \log_2 e} = 2^{i+d} = 2^i \times 2^d, x \in (-\infty, +\infty), d \in [0, 1) \tag{7}$$

where $i$ and $d$ represent the integer part and fractional part of $x \log_2 e$. The IEEE-754 standard states that with a known floating-point number $f$, the calculation of $f \times 2^i$ can be accomplished by adding the exponent of $f$.

The expression of the logarithmic function implemented in this paper is Equation (8).

$$y = \ln x, x \in (0, +\infty) \tag{8}$$

According to the IEEE-754 standard, Equation (8) can be transformed into Equation (9):

$$y = \ln x = \ln \left(2^{E-127} \times (1 + \frac{T}{2^{23}})\right) = (E - 127) \times \ln 2 + \ln \left(1 + \frac{T}{2^{23}}\right) \tag{9}$$

where $E$ and $T$ denote the exponent part and mantissa part of x, respectively, and $1 + \frac{T}{2^{23}}$ falls within the range $[1, 2)$.

The domains and codomain of the transcendental function and the core computing unit after a preliminary range reduction are shown in Table 1.

**Table 1.** The domains and codomain of the transcendental function and the core computing unit.

| Transcendental Function | Domains | Codomain | Core Computing Unit | Inputs | Outputs |
|---|---|---|---|---|---|
| $y = \sin \pi x$ | $(-2^{128}, -2^{-126}] \cup [2^{-126}, 2^{128})$ | $[-1, 1]$ | $\hat{y} = \sin \pi \hat{x}$ | $[0, 0.5]$ | $[0, 1]$ |
| $y = \cos \pi x$ | $(-2^{128}, -2^{-126}] \cup [2^{-126}, 2^{128})$ | $[-1, 1]$ | $\hat{y} = \sin \pi \hat{x}$ | $[0, 0.5]$ | $[0, 1]$ |
| $y = \frac{\arctan(x)}{\pi}$ | $(-2^{128}, -2^{-126}] \cup [2^{-126}, 2^{128})$ | $(-0.5, 0.5)$ | $\hat{y} = \frac{\arctan(\hat{x})}{\pi}$ | $[0, 1]$ | $[0, 0.25]$ |
| $y = e^x$ | $[-126ln2, 128ln2)$ | $[2^{-126}, 2^{128})$ | $\hat{y} = 2^{\hat{x}}$ | $[0, 1)$ | $[1, 2)$ |
| $y = \ln x$ | $[2^{-126}, 2^{128})$ | $[-126ln2, 128ln2)$ | $\hat{y} = \ln \hat{x}$ | $[1, 2)$ | $[0, \ln 2)$ |

*3.2. Polynomial Fitting*

Each function in Table 1 is a nonlinear function, and a segmented polynomial fit is performed in order to obtain high-precision results. The higher level of polynomial fitting brings a higher accuracy in the operation results, but also more basic operation units [33]. Each higher order of polynomial requires one more adder and one more multiplier, and the data path delay will be longer at the same time [34].

Polynomials can be calculated in two ways, namely, floating point and fixed point [35,36]. A floating-point operation means that the entire operation is performed in floating-point form. A fixed-point operation means that the operation is performed in fixed-point form, while the inputs and outputs are in floating-point form. Floating-point arithmetic is simple in design and obtains a high computational accuracy, but it brings greater side effects such as a large latency and a large area. On the premise of ensuring the calculation accuracy [37], a fixed-point operation has a smaller latency and area.

In order to improve the area and performance of the hardware design, this paper used a fixed-point binomial operation. The specific implementation steps are shown below, and the flow chart of the methodology is shown in Figure 1.

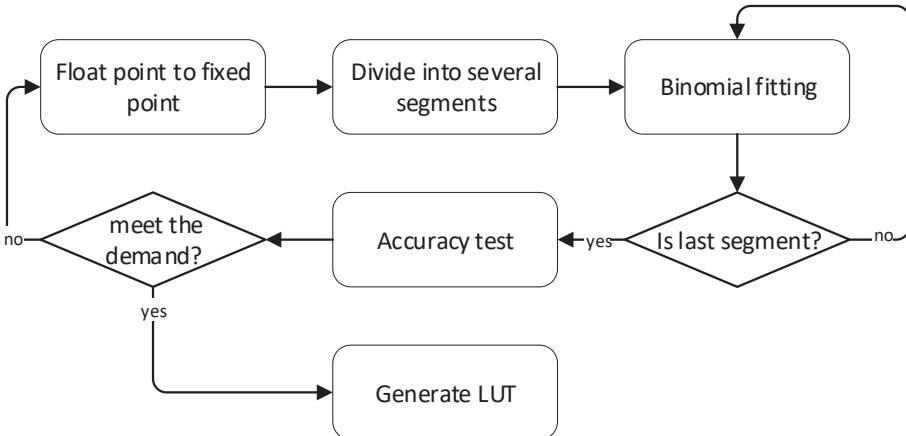

**Figure 1.** The flow chart of the methodology.

1. Transform the inputs and corresponding theoretical outputs into fixed-point form.
2. Divide the input range evenly into several segments.
3. Each segment is fitted with a binomial to obtain binomial coefficients (*a*, *b* and *c*), and the binomial coefficients are in fixed form and stored in a lookup table. The address of the lookup table comes from the high bits of the input, and the low bits of the input and corresponding output are used to perform the binomial fit of the segment. The process of binomial fitting is shown in Figure 2.

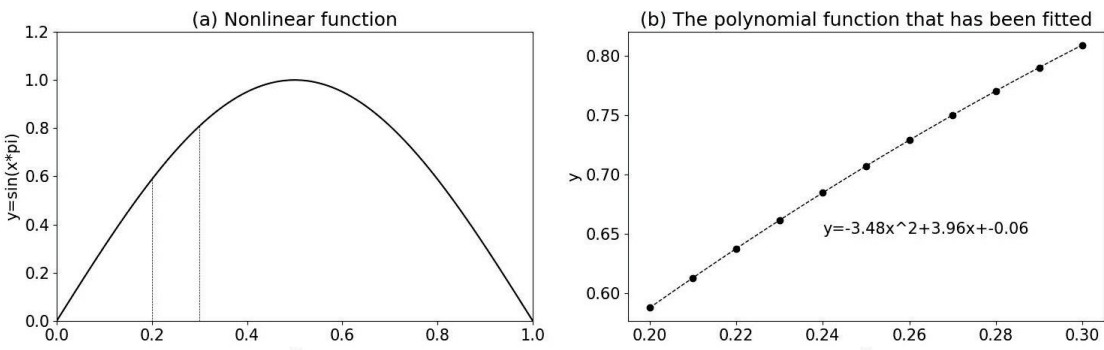

**Figure 2.** The process of binomial fitting. (**a**) Nonlinear function and the selected segment. (**b**) Fitting curve obtained by binomial fitting according to the scatter point.

4. Run the operation of Equation (10) and perform accuracy tests to count the maximum absolute error of the core computing unit.

$$y = ax^2 + bx + c \qquad (10)$$

The testing process requires determining the data's bit width of the core, including that of the two multipliers and the two adders. In order to get a high accuracy, the data's bit width of the core computing unit should be consistent with the data's bit width of the input and the binomial coefficients.

5. If the accuracy does not meet the demand, reset the above parameters for segmented binomial fitting and test again until the accuracy meets the demand. Finally, generate the lookup table.

In the process of configuration, the fixed-pointing parameters, the number of segments and the data's bit width of the core computing unit all affect the accuracy of the final results. In order to obtain a high-precision transcendental function binomial coefficient lookup table, the control variable method was used to obtain the most suitable configuration parameters for the binomial fitting.

Firstly, the data's bit width of the core operation unit was set as large as possible to ensure that no large error would be introduced in the binomial operation, and then different segment numbers and different fixed-point parameters were set to compare the maximum error of the results of each transcendental function. The relationship between the logarithm based on 2 of the maximum error and the number of segments, the fixed-point parameter of each transcendental function is shown in the Figure 3. When the number of segments was 32 or 64, it was difficult to meet the high-accuracy requirements regardless of the fixed-point parameters. When the number of segments was 128 or 256, there was no significant difference in the accuracy of the two conditions when the degree of fixed-pointing was low, and as the degree of fixed-pointing increased, the accuracy under the configuration of 256 segments was slowly higher than that of 128 segments. Correspondingly, the higher the number of parameters, the larger the lookup table [38,39]. In order to get the most suitable configuration combining precision and area, the number of segments in this paper was determined to be 128 and the fixed-point parameter was 26.

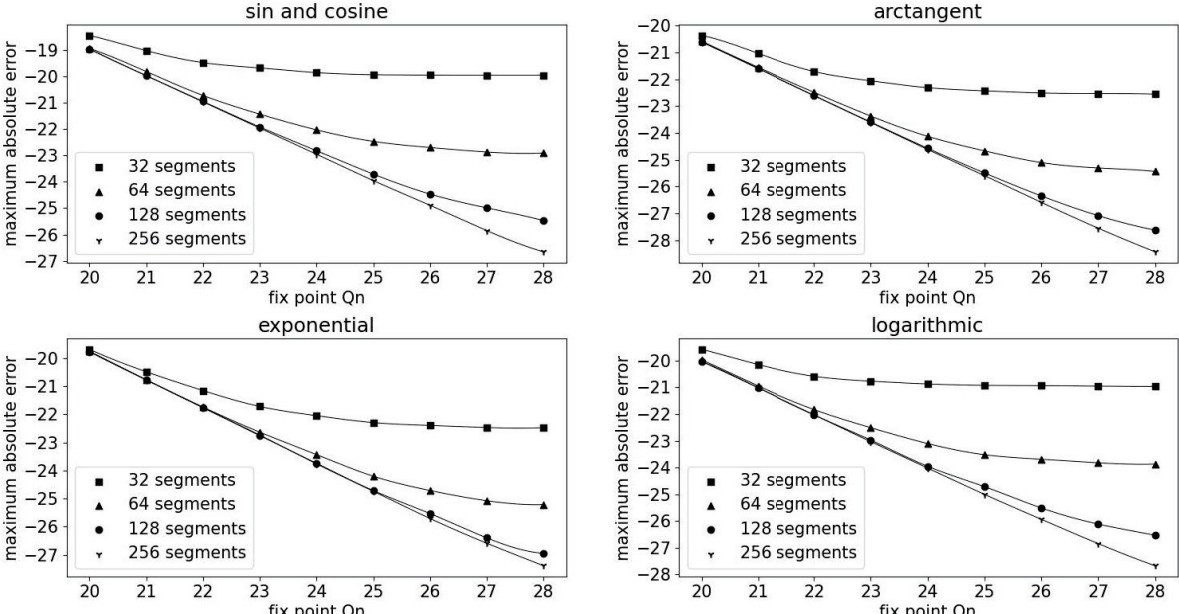

**Figure 3.** The relationship between the maximum error and the number of segments, the fixed-point parameter.

Then, the data's bit width of the core computing unit was optimized for reduction, which led to an overall upward shift of the maximum error curve. The degree of errors introduced in different data's bit width configurations was different, so we experimented with each data's bit width configuration, selected a few better configurations and compared them together, as shown in Figure 4 and Table 2. According to the figure, it can be concluded that the sine, cosine, arctangent, exponential and logarithmic functions could achieve the highest accuracy, respectively, in the cases of configuration 5, configuration 3, configuration 3 and configuration 6.

**Table 2.** The normalized value of the maximum absolute error of functions under different configurations.

| Functions | cfg1 | cfg2 | cfg3 | cfg4 | cfg5 | cfg6 | cfg7 | cfg8 |
|---|---|---|---|---|---|---|---|---|
| Sine and cosine | −23.60 | −23.88 | −23.60 | −23.74 | −24.18 | −23.95 | −23.03 | −23.74 |
| Arctangent | −25.42 | −25.48 | −26.13 | −25.45 | −25.52 | −25.95 | −24.89 | −25.45 |
| Exponential | −23.75 | −24.21 | −24.45 | −23.37 | −23.74 | −24.29 | −22.86 | −23.37 |
| Logarithmic | −24.91 | −24.72 | −24.98 | −24.93 | −24.96 | −25.06 | −24.36 | −24.84 |

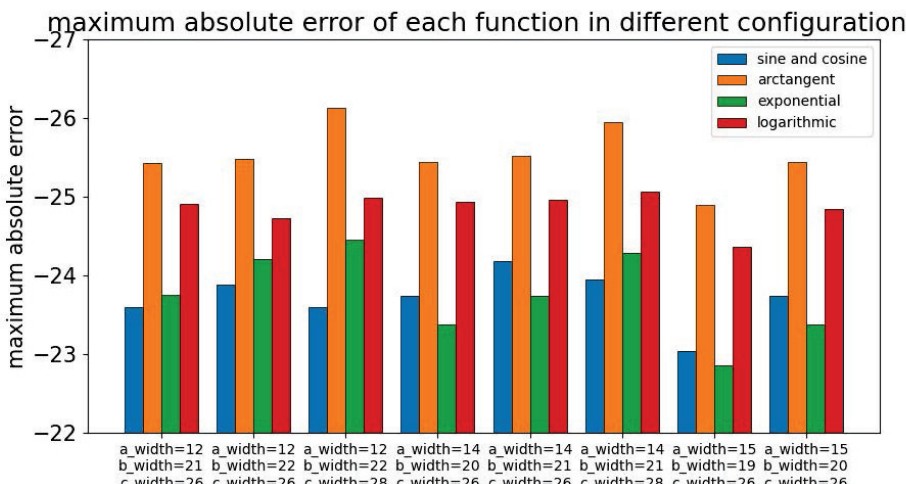

**Figure 4.** The relationship between the maximum error and the different configurations.

The size of the lookup table under each configuration is shown in Figure 5. Combining the accuracy and the lookup table size of each function under each configuration, this paper used configuration 2 for the hardware design, which had binomial coefficients (*a*, *b* and *c*) in the lookup table of 12 bits, 22 bits and 26 bits, because the maximum absolute error of all functions and hardware resources were relatively balanced under this configuration.

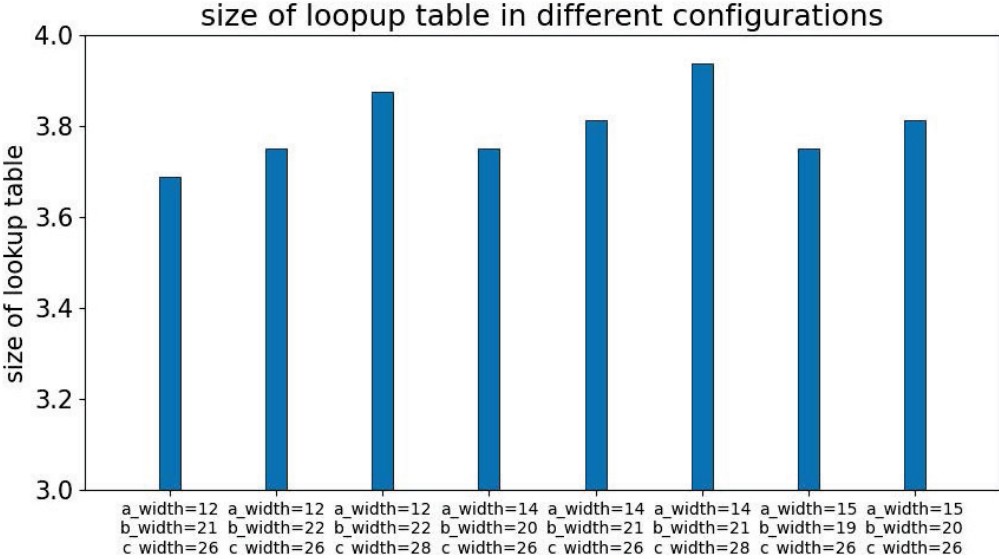

**Figure 5.** The relationship between the size of the lookup table and the different configurations.

### 3.3. Hardware Implementation

The reconfigured hardware architecture proposed in this paper depended on the fact that all functions utilized the same core computing unit. There were also preprocessing and post-processing modules besides the core computing unit, and these modules are further explained in later sections. The whole architecture is shown in Figure 6.

The preprocessing module was used for reducing the input range, fixed-pointing and providing the sideband signal required for the postprocessing of each function. Then, the core module performed the corresponding binomial operations. At last, the postprocessing module processed the sideband signal, turned the results into a floating-point form and output them.

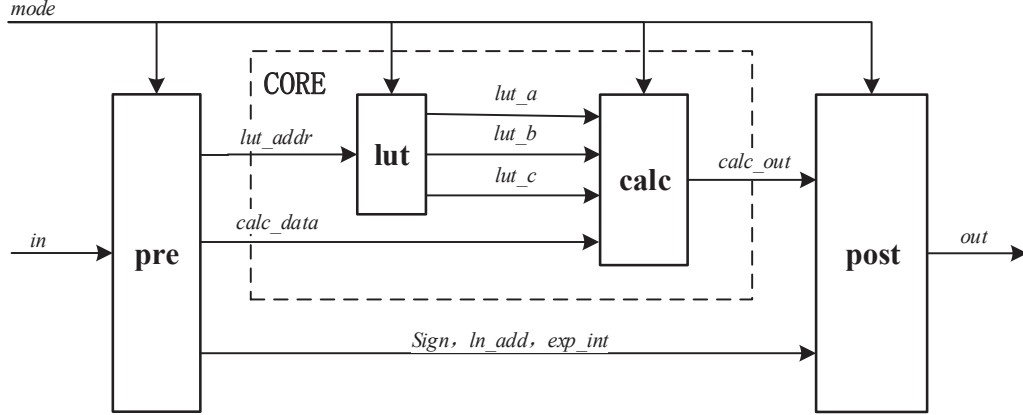

**Figure 6.** Floating-point transcendental function hardware architecture, where *pre* represents the preprocessing module, *lut* represents the lookup table module, *calc* represents the calculation module and *post* represents the postprocessing module.

### 3.3.1. Reconfigurable Hardware Architecture

The hardware architecture was reconfigurable because of the high reuse of hardware resources. That is, all supported functions shared the same core computing unit, which meant that the more transcendent functions were supported, the higher the hardware resource utilization. In addition, the difference between data paths for different functions only existed in the preprocessing and postprocessing stages, which were also reconfigurable. In these two stages, the calculation of different functions reused the available hardware resources as much as possible, and there was no combinatorial logic loop.

### 3.3.2. Preprocessing Module

The preprocessing module is shown in Figure 7. It receives floating-point data (*in*) and function mode (*mode*) as inputs and outputs signal bit (*sign*), the address of the LUT (*lut_addr*), calculates data (*calc_data*), the compensation value of exponent (*exp_int*) and the compensation value of logarithm (*ln_add*), where *exp_int* represents $2^i$ in Equation (7) and *ln_add* represents $(E - 127) \times \ln 2$ in Equation (9). The preprocessing module includes a floating-point-to-fixed-point module (*float2int*), a data inversion module (*inv*), a data inversion selection module (*inv_sel*), an 8-bit subtracter, a 34-bit integer multiplier and several multiplexers.

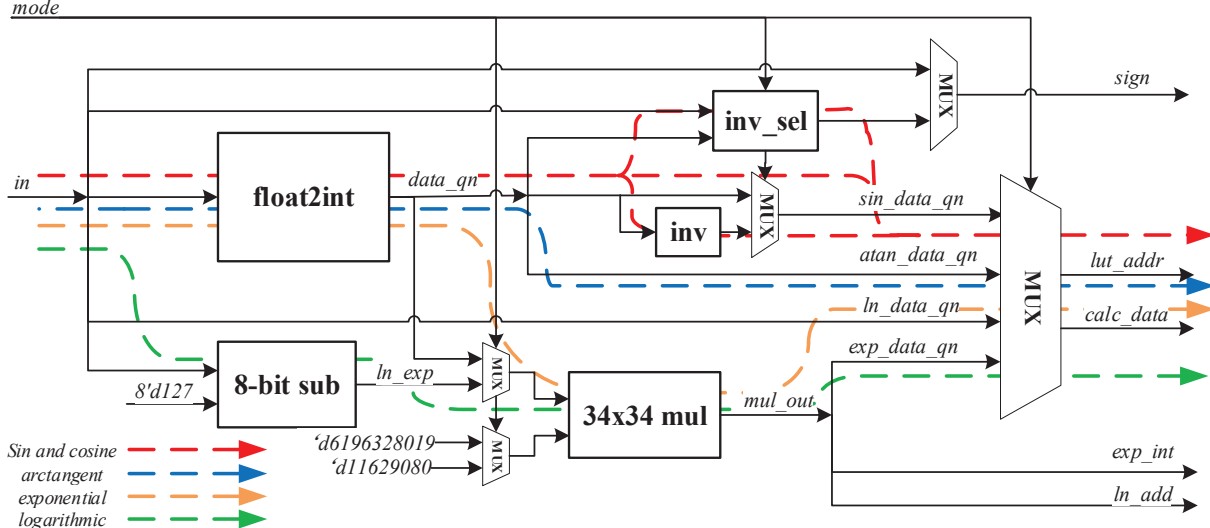

**Figure 7.** Preprocessing module.

The data pass through *float2int*, *inv* and *inv_sel* when operating sine and cosine functions and passes through *float2int* when the arctangent function is operated. If an exponent function is expected to be calculated, *float2int* and the 34-bit integer multiplier, whose input of 6,196,328,019 is the $Q_{32}$ fixed-point number of $\log_2 e$, are used. If we choose to operate a logarithmic function, the data pass through the 8-bit subtracter and the 34-bit multiplier successively, whose input of 11,629,080 is the $Q_{24}$ fixed-point number of $\ln 2$.

### 3.3.3. Core Computing Unit

The lookup table module is shown in Figure 8. It can output the binomial coefficients based on the address and operation mode.

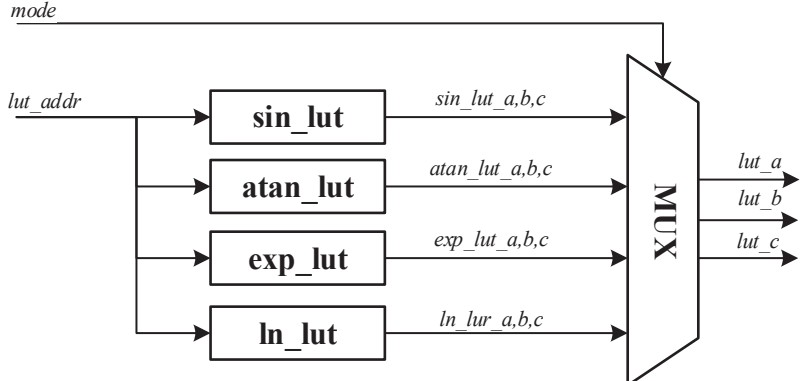

**Figure 8.** Lookup table module.

It contains four binomial-fitting coefficient tables in Table 1. Each table contains 128 sets of coefficients. The coefficients are 12 bits, 22 bits and 26 bits, respectively. The total storage space of these tables is 3.75 KB.

The calculation module is shown in Figure 9.

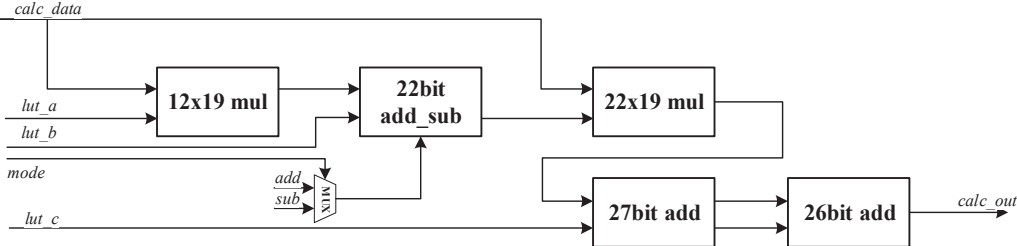

**Figure 9.** Calculation module.

The calculation module outputs the result (*calc_out*) based on the calculating data (*calc_data*), binomial coefficients (*lut_a*, *lut_b*, *lut_c*) and the operation mode (*mode*). It is composed of two multipliers, two adders and one adder/subtracter to calculate an output with an input in Equation (11). Compared with Equation (10), this calculation method has one fewer multiplier and a lower latency.

$$calc\_out = (lut\_a \times calc\_data + lut\_b) \times calc\_data + lut\_c \tag{11}$$

The 12 × 19-bit integer multiplier calculates $lut\_a \times calc\_data$. Then, the 22-bit adder/subtracter outputs $lut\_a \times calc\_data + lut\_b$. Since the coefficients $lut\_b$ in the sine and cosine functions are negative and are positive otherwise, the adder/subtracter performs a subtraction in the sine and cosine functions and an addition otherwise. Then, after the operation of the 22 × 19-bit integer multiplier and 27-bit adder, we can get the initial result $(lut\_a \times calc\_data + lut\_b) \times calc\_data + lut\_c$. Finally, the 26-bit adder performs a rounding and we can get a fixed-point Q26 output result.

### 3.3.4. Postprocessing Module

The postprocessing module is shown in Figure 10. It can output floating-point results (*out*) based on (*calc_out*), signal bit (*sign*), the compensation value of exponent (*exp_int*) and the compensation value of logarithm (*ln_add*). It consists of a fixed-point-to-floating-point module (*int2float*), a sign-processing module (*sign_proc*), a 33-bit adder, an 8-bit adder and several multiplexers.

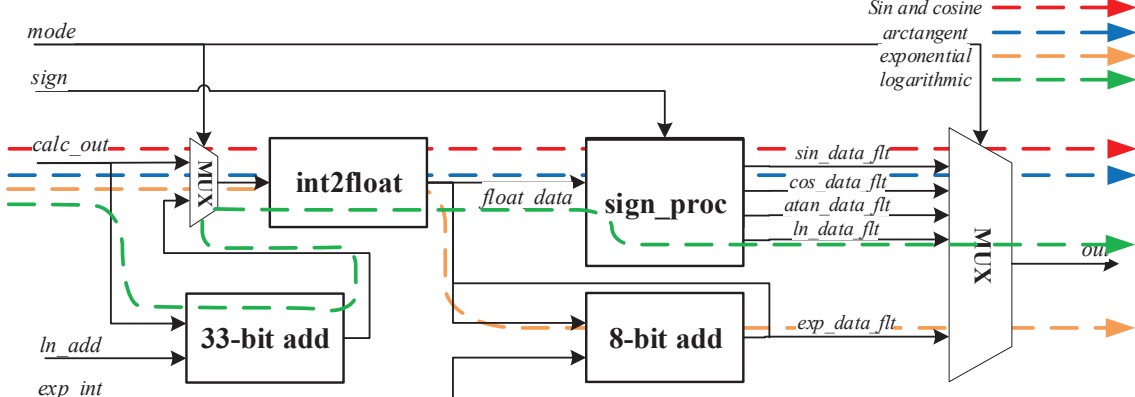

**Figure 10.** Postprocessing module.

The data pass through *int2float* and *sign_proc* when operating the sine, cosine and arctangent functions. If an exponent function is expected to be calculated, *int2float* and the 8-bit adder are employed, where the 8-bit adder is used to add *exp_int* to the output of *int2float* to obtain the real output value. If we choose to operate a logarithmic function, the 33-bit adder, *int2float* and *sign_proc* are employed, where the 33-bit adder is used to add *ln_add* to *calc_out*.

### 3.3.5. Retiming Optimization

To support the pipeline architecture, we added several stages of pipeline registers in the end module, as shown in Figure 11. Then, we set the command *set_optimize_registers* for the retiming optimization to properly move pipeline registers, which aimed to ensure that the combined logic delays between both stages were essentially the same. The more pipeline registers there were, the higher the frequency the hardware circuit could work at, but the more cycles it took to compute the transcendence function.

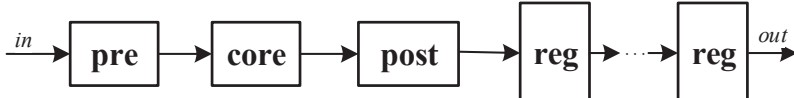

**Figure 11.** Pipelined architecture of hardware by adding several registers after the original hardware design, where *pre* represents the preprocessing module, *core* represents the core computing unit, *post* represents the postprocessing module and *reg* represents the register.

Different transcendental functions work with different data paths. Sine, cosine and arctangent functions have shorter data paths in the preprocessing and postprocessing modules, while the exponential and logarithmic functions have longer ones. The above method of designing all functions with the same pipeline structure has performance losses for the transcendental functions with a shorter data path. As a result, the pipeline registers must be arranged according to the data path of the transcendent function before using the command *set_optimize_registers* for the retiming optimization, as shown in Figure 12, where *path*0 represents a shorter data path and *path*1 represents a longer one. After optimization, with the same hardware circuit, the sine, cosine and arctangent functions with shorter data paths will spend fewer clock cycles than the exponential and logarithmic functions with longer data paths, which can improve performance.

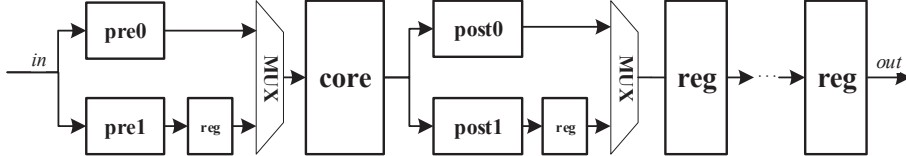

**Figure 12.** Optimizing the pipeline architecture of the hardware, where *pre* represents the preprocessing module, *core* represents the core computing unit, *post* represents the postprocessing module, *reg* represents the register, *Mux* represents multiplexers.

## 4. Experiment and Comparison

### 4.1. Accuracy

The test method was to traverse all normalized floating-point numbers and compare the circuit calculation results with the results of the C language math library. The hardware accuracy test results are shown in Table 3.

**Table 3.** Hardware accuracy.

| Functions | 0 $ulp$ | 1 $ulp$ | 2 $ulp$ |
|---|---|---|---|
| Sine | 77.065% | 13.719% | 9.216% |
| Cosine | 77.065% | 13.719% | 9.216% |
| Arctangent | 78.896% | 10.669% | 10.435% |
| Exponential | 61.416% | 38.546% | 0.038% |
| Logarithmic | 75.935% | 11.376% | 12.689% |

The precision of the floating-point transcendental function hardware algorithm proposed in this paper, according to the data in Table 3, could satisfy 2 $ulp$, that is, the maximum error of the binary representation of the single-precision floating-point number between the hardware output result and the C language math library result was two.

### 4.2. Performance, Power and Area

First, we synthesized the hardware circuit that supported all transcendental functions, which was implemented by directly inserting pipeline registers, using Design Compiler under the UMC 40 nm CMOS process library, 0.99 V, 125 °C and RVT conditions. The results of the performance, power and area are shown in the Table 4.

**Table 4.** Performance, power and area of a hardware circuit which supports all transcendental functions.

| Pipelines | Frequence/MHZ | Area/$\mu m^2$ | Latency/ns | Power/mW |
|---|---|---|---|---|
| 2 | 100 | 13,907.42 | 20.00 | 0.274 |
| 3 | 170 | 14,414.78 | 17.65 | 0.453 |
| 4 | 230 | 15,062.91 | 17.39 | 1.234 |
| 5 | 290 | 16,852.14 | 17.24 | 1.084 |
| 6 | 340 | 16,191.11 | 17.64 | 1.567 |
| 7 | 380 | 15,745.48 | 18.42 | 2.618 |

As we can see, the floating-point transcendental function hardware circuit could achieve a high performance of only 17.39 ns with fewer hardware resources when it worked at a frequency of 230 MHz and cost four cycles. The full-load power consumption was 1.234 mW, and the area was 15,062.91 $\mu m^2$. The area of each module was as shown in Figure 13. As can be seen from Figure 13, because of the reconfigurable technology, the area of the core calculation module only took up 28.18% of the total area.

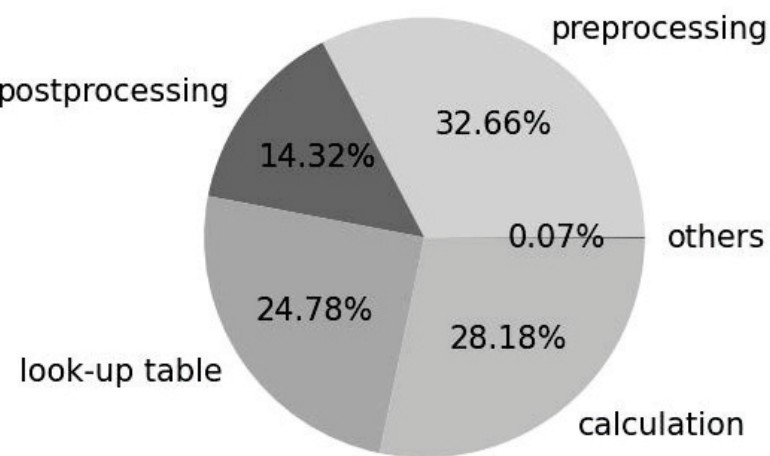

**Figure 13.** The area ratio of each module.

Then, we synthesized the hardware circuits that only enabled one transcendental function separately, under the same conditions, and set the frequency to 230 MHz. The performance, power and area results of each function are shown in the Table 5. The data in the Table 5 show that when different functions were implemented separately at the same frequency, the number of cycles required was different. Among them, the cycle of the sine, cosine and arctangent functions was shorter than that of the exponential and logarithmic functions.

**Table 5.** Performance, power and area of a hardware circuit which supports a single transcendental function.

| Functions | Pipelines | Area/$\mu m^2$ | Latency/ns | Power/mW |
|-----------|-----------|-----------------|------------|----------|
| Sine | 3 | 6802.79 | 13.04 | 0.3132 |
| Cosine | 3 | 6802.79 | 13.04 | 0.3132 |
| Arctangent | 3 | 6187.03 | 13.04 | 0.3725 |
| Exponential | 4 | 7976.67 | 17.39 | 0.6075 |
| Logarithmic | 4 | 5968.74 | 17.39 | 0.2630 |

For comparison, the hardware circuit that was created by the direct combination of the hardware circuit that independently realized the transcendental function had an area and power consumption of 26,935.23 $\mu m^2$ and 1.511 mW, respectively. Compared with the hardware circuit without reconfigurable technology, the area and power consumption of the proposed reconfigurable hardware architecture were reduced by 44.08% and 18.33%, respectively.

Finally, we synthesized a hardware circuit after pipeline register optimization, which supported the calculation of the sine, cosine and arctangent functions in three cycles and the calculation of the exponential and logarithmic functions in four cycles. The results compared with those of the hardware circuits supporting all functions with four cycles are shown in the Table 6. The results show that the highest working frequency became lower, and the working performance for the exponential and logarithmic functions was degraded by 4.35%. However, the performance for the sine, cosine and arctangent functions was improved by 21.59% and the area and power consumption were both improved.

**Table 6.** Hardware comparison before and after optimization.

| Hardware Circuit | Max Frequency/MHz | Area/μm² | Latency/ns | Power/mW |
|---|---|---|---|---|
| Before optimization | 230 | 15,062.91 | 17.39 | 1.234 |
| After optimization | 220 | 14,009.35 | 18.18/13.63 | 0.923 |
| | −4.35% | −7.00% | +4.35%/−21.59% | −25.20% |

As shown in Table 7, the area and power consumption of the hardware circuit after pipeline register optimization were reduced by 47.99% and 38.91%, respectively, when compared to the hardware circuit without reconfigurable technology. The proposed hardware supported five different transcendental function operations, and its performance was improved by 1.3 times and the area was reduced by 27.2% compared with those of [37]. In addition, the accuracy was improved by 75% compared with that in [26]. Compared with state-of-the-art hardware accelerators for transcendental functions, the reconfigurable hardware architecture proposed in this paper had certain advantages in performance, accuracy and area, which can be seen in Table 8.

**Table 7.** Comparison with the hardware circuit without reconfigurable technology.

| Hardware Circuit | Max Frequence/MHz | Area/μm² | Latency/ns | Power/mW |
|---|---|---|---|---|
| Separate implementation | 230 | 26935.23 | 17.39 | 1.511 |
| After optimization | 220 | 14009.35 | 18.18/13.63 | 0.923 |
| | −4.35% | −47.99% | +4.35%/−21.59% | −38.91% |

**Table 8.** Comparison of results with state-of-the-art hardware architectures.

| Paper | This | [26] | [36] | [37] | [27] |
|---|---|---|---|---|---|
| Function | All | Logarithmic | Sine, cosine | exp | All |
| Technique | UMC 40 nm | 65 nm | FPGA | STM 65 nm | − |
| Accuracy | 2 *ulp* | 3.5 *ulp* | 1.5 *ulp* | $1 \times 10^{-9}$ | − |
| Performance | 17.39 *ns* | 96 *ns* | 15 cycles | 40.3 *ns* | 14.5 *τ* |
| Power | 1.234 mW | − | − | 0.959 mW | − |
| Area | 15,062 | 60,000 gate | − | 20,700 | − |

## 5. Conclusions

In order to support multiple floating-point transcendental function operations with a small hardware circuit area, this paper proposed a reconfigurable hardware architecture for miscellaneous floating-point transcendental functions. This paper utilized a reconfigurable technology to implement multiple transcendental functions, including sine, cosine, arctangent, exponential and logarithmic functions. The transcendental function hardware accelerator with a high accuracy and low latency, which is significant for many application scenarios, cost a small quantity of hardware resources. In this paper, the method of combining lookup tables with binomial operations, which generated lookup tables occupying 3.75 KB of space, was used to design a hardware accelerator of high-precision transcendental functions.

The experimental results showed that the difference between the calculation results of the proposed hardware circuit and those of the C language math library was at most 2 *ulp*. Under the UMC 40 nm CMOS process library, the clock frequency could reach 220 MHz with a latency of 18.18 ns, a full-load power consumption of 0.923 mW and an area of $1.40 \times 10^4$ μm². Compared with five separate superfunction hardware accelerators, the area

was reduced by 47.99% and the power was reduced by 38.91%. In some area-sensitive application scenarios that require a low latency and a high precision for transcendental function operations, the floating-point transcendental function hardware architecture proposed in this paper has important application value. Moreover, the reconfigurable architecture proposed in this paper will play an even greater role in the future as various fields pursue high-performance computing.

**Author Contributions:** Conceptualization: P.L. and H.J.; methodology: P.L., H.J. and W.X.; validation: H.Y. and C.X.; formal analysis: K.H.; investigation: P.L. and W.X.; data curation: H.Y. and C.X.; writing—original draft preparation: H.J.; writing—modified and polished: P.L. and K.H.; supervision: K.H. All authors have read and agreed to the published version of the manuscript.

**Funding:** This work is funded by the National Key R&D Program of China (2020YFB0906000, 2020YFB0906001).

**Institutional Review Board Statement:** Not applicable.

**Informed Consent Statement:** Not applicable.

**Data Availability Statement:** Not applicable.

**Acknowledgments:** Many thanks to editors and reviewers for their comments and help.

**Conflicts of Interest:** The authors declare no conflict of interest.

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
