# Peer review of "A Reconfigurable Hardware Architecture for Miscellaneous Floating-Point Transcendental Functions"

_electronics, doi:10.3390/electronics12010233_

Round 1

Reviewer 1 Report

The paper states the importance of their results as transcendental functions are indeed very useful in several domains of human endeavor. Nevertheless, the state-of-the-art needs to be explained further to sustain the importance of the breakthrough that was achieved by the authors.

The authors are right when they explain that other hardware accelerators that are available support few functions or demand lots of resources, while their proposal is flexible (allows a greater number of functions) and resource-efficient (in terms of area, power, and processing times). However, the authors need to clearly explain these advantages in depth throughout the paper, especially on the conclusions/discussion sections.

The paper does not provide an adequate definition for important terms like reconfigurable hardware, binomial fitting, look-up table, power, accuracy, performance, area, etc. This would allow unexperienced readers or readers from other fields to understand the paper and its important contributions.

In general, the text that is presented on the figures is hard to read. Make sure that the size of the text is adequate. Another observation regarding the figures is that the footer text should be self-contained, even if the figures are explained somewhere else, the figure text should explain clearly what the figure is about and why it is presented.

There are small grammar, spelling, and style problems throughout the paper, please correct them. Please use the following LaTex package https://ctan.org/pkg/siunitx to help in the formatting of physical units. A PDF file is provided to help the authors identify some specific problems on the manuscript.

Author Response

Thank you for your valuable revision suggestions. Here is our response to your comments. Please see the attachment.

Reviewer 2 Report

The article is on a very good level.

I have only a few comments.

Regarding Figures number 1, there is a text, which is too small and hard to read. And the Figures are unnecessarily small as well.

In the bottom part of Figure No. 2 and No. 3, there is yet again a small illegible text.

A text enlargement would be beneficial in Figure No. 4 as well.

I recommend indenting the first word on line number 202 by a space. Also the word "sine" and "cosine", on this very line, should be separated by adding a space.

When it comes to Figures No. 5, 7, and 8, I advise increasing the size to achieve a larger text.

There is an excessive dot on line number 240.

Author Response

(The authors gave the same response as above.)
